# FACTORIZED IMPLICIT GLOBAL CONVOLUTION FOR AUTOMOTIVE COMPUTATIONAL FLUID DYNAMICS PREDICTION

## ABSTRACT

Computational Fluid Dynamics (CFD) is crucial for automotive design, requiring the analysis of large 3D point clouds to study how vehicle geometry affects pressure fields and drag forces. However, existing deep learning approaches for CFD struggle with the computational complexity of processing high-resolution 3D data. We propose Factorized Implicit Global Convolution (FIGConv), a novel architecture that efficiently solves CFD problems for very large 3D meshes with arbitrary input and output geometries. FIGConv achieves quadratic complexity $O(N^2)$, a significant improvement over existing 3D neural CFD models that require cubic complexity $O(N^3)$. Our approach combines Factorized Implicit Grids to approximate high-resolution domains, efficient global convolutions through 2D reparameterization, and a U-shaped architecture for effective information gathering and integration. We validate our approach on the industry-standard Ahmed body dataset and the large-scale DrivAerNet dataset. On DrivAerNet, our model achieves an $R^2$ value of 0.95 for drag prediction, outperforming the previous state-of-the-art by a significant margin. This represents a 40% improvement in relative mean squared error and a 70% improvement in absolute mean squared error over prior methods.

## 1 INTRODUCTION

The automotive industry stands at the forefront of technological advancement, relying heavily on computational fluid dynamics (CFD) to optimize vehicle designs for enhanced aerodynamics and fuel efficiency. The accurate simulation of complex fluid dynamics around automotive geometries is crucial for achieving optimal performance. However, traditional numerical solvers, including finite difference and finite element methods, often prove computationally intensive and time-consuming, particularly when dealing with large-scale simulations, as encountered in CFD applications. The demand for efficient solutions in the automotive sector necessitates the exploration of innovative approaches to accelerate fluid dynamics simulations and overcome the limitations of current solvers.

In recent years, deep learning methodologies have emerged as promising tools in scientific computing, advancing traditional simulation techniques, in bio-chemistry (Jumper et al., 2021), seismology (Yang et al., 2021), climate change mitigation (Wen et al., 2023), and weather (Pathak et al., 2022; Lam et al., 2022) to name a few. In fluid dynamics, recent attempts develop domain specific deep learning methods to emulate fluid flow evolution on 2D and 3D proof of concept settings (Jacob et al., 2021; Li et al., 2020b; Pfaff et al., 2020a; Kossaifi et al., 2023). While most of these works focused on solving problems using relatively low-resolution grids, industrial automotive CFD requires working with detailed meshes containing millions of points.

To address the time-consuming and compute-intensive nature of conventional CFD solvers on detailed meshes, recent studies (Jacob et al., 2021; Li et al., 2023) have explored replacing CFD simulations with deep learning-based models to accelerate the process. In particular, Jacob et al. (2021) studies DrivAer dataset (Heft et al., 2012), utilize Unet (Ronneberger et al., 2015) architecture and aim to predict single number scalar car surface drag coefficients – the integration of surface pressure and friction – directly by bypassing the integration. Furthermore, the architecture is applied on 3D voxel grids that requires $O(N^3)$ complexity, forcing the method to scale only to low-resolution 3D grids. Li et al. (2023) propose a neural operator method for Ahmed body (Ahmed et al., 1984) car dataset and aims to predict the pressure function on the car surface. This approach utilizes graph

embedding to a uniform grid and perform 3D global convolution through fast Fourier transform (FFT). While this method, in principle, handles different girding, the FFT in the operator imposes complexity of $O(N^3 \log N^3)$, which becomes computationally prohibitive as the size of the grid increases. Both methods face scalability challenges due to their cubic complexity, which severely limits their representational power for high-resolution simulations. Consequently, there is a pressing need for a specialized, domain-inspired method capable of handling 3D fine-grained car geometries with meshes comprising tens of millions of vertices (Jacob et al., 2021). Such massive datasets demand a novel approach in both design and implementation.

**In this work**, we propose a novel neural CFD approach with quadratic complexity $O(N^2)$, significantly improving scalability over existing 3D neural CFD models that require cubic complexity $O(N^3)$. Our method outperforms the state-of-the-art by reducing absolute mean squared error by 70%.

The key innovations of our approach include Factorized Implicit Grids and Factorized Implicit Convolution. With Factorized Implicit Grids, we approximate high-resolution domains using a set of implicit grids, each with one lower-resolution axis. For instance, a $1k \times 1k \times 1k$ domain containing $10^9$ elements can be represented by three implicit grids with dimensions $5 \times 1k \times 1k$, $1k \times 4 \times 1k$, and $1k \times 1k \times 3$. This reduces the total elements to just $5M + 4M + 3M = 12M$, a significant reduction from the original $10^9$. Our Factorized Implicit Convolution method approximates 3D convolutions using these implicit grids, employing reparametrization techniques to accelerate computations.

We validate our approach on two large-scale CFD datasets: DrivAerNet (Heft et al., 2012; Elrefaie et al., 2024) and Ahmed body dataset (Ahmed et al., 1984). Our experiments focus on surface pressure and drag coefficient prediction. Results demonstrate that our network is an order of magnitude faster than existing methods while achieving state-of-the-art performance in both drag coefficient prediction and per-face pressure prediction.

## 2 RELATED WORK

The integration of deep learning into CFD processes has seen significant research efforts. Graph neural operator is among the first methods to explore neural operators on various geometries and meshes (Li et al., 2020b). The architectures based on graph neural networks (Ummenhofer et al., 2019; Sanchez-Gonzalez et al., 2020; Pfaff et al., 2020a), follow message passing and encounter similar computational challenges when dealing with realistic receptive fields. The u-shaped graph kernel, inspired by multipole methods and UNet (Ronneberger et al., 2015), offers an innovative approach to graph and operator learning (Li et al., 2020c). However, the core computational challenges in 3D convolution remain nonetheless, even for FNO based architectures that are widely deployed (Li et al., 2022; Pathak et al., 2022; Wen et al., 2023). Deep learning models in computer vision, e.g., UNet, have been used to predict the fluid average properties, such as final drag for the automotive industry (Jacob et al., 2021; Trinh et al., 2024). Studies incorporating signed distance functions (SDF) to represent geometry have gained attention where CNNs are used as predictive models in CFD simulations (Guo et al., 2016; Bhatnagar et al., 2019). The 3D representation of SDF inflicts significant computation costs on the 3D models, making them only scale to low-resolution SDF, missing the details in the fine car geometries. Beyond partial differential equations (PDE) and scientific computing, various deep learning models have been developed to deal with fine-detail 3D scenes and objects. In particular, for dense prediction tasks in 3D space, a network is tasked to make predictions for all voxels or points, for which 3D UNets have been widely used for, e.g., segmentation (Li et al., 2018; Atzmon et al., 2018; Hermosilla et al., 2018; Graham and van der Maaten, 2017; Choy et al., 2019). However, many of these networks exhibit poor scalability due to the cubic complexity of memory and compute $O(N^3)$ or slow neighbor search.

Recently, decomposed representations for 3D – where multiple orthogonal 2D planes have been used to reconstruct 3D representation – have gained popularity due to their efficient representation and have been used in generation (Chan et al., 2022; Shue et al., 2023) and reconstruction (Chen et al., 2022; Fridovich-Keil et al., 2023; Cao and Johnson, 2023). Such representation significantly reduces the memory complexity of implicit neural networks on 3D continuous planes. Despite basing on the decomposition of continuous planes and fitting a single neural network to a scene, this approach shares relevance with our factorized grid convolution approach.

Prior works in the deep learning literature, focusing on large-scale point clouds, ranges from the use of graph neural networks and point-nets to the u-shaped architectures along with advanced neighborhood search (Qi et al., 2017a; Hamilton et al., 2017; Wang et al., 2019; Choy et al., 2019; Shi et al., 2020). However, these methods make assumptions that may not be valid when applied to CFD problems. For example, the sub-sampling approach is a prominent approach to deal with the social network, classification, and segmentation to gain robustness and accuracy. However, in the automotive industry, dropping points could lead to a loss of fine-details in the geometry, the vital component of fluid dynamics evolution and car design. There is a need for a dedicated domain-inspired method able to work directly on fine-grained car geometry with meshes composed of $100M$ vertices (Jacob et al., 2021), a massive size that requires a unique design and treatment.

## 2.1 FACTORIZATION

Factorization of weights in neural networks has been studied to reduce the computational complexity of deep learning models Panagakis et al. (2021). It has been applied to various layers, including full-connected Novikov et al. (2015), and most recently, the low-rank adaptation of transformers (Hu et al., 2021), and the training of neural operators (Kossaifi et al., 2024). In the context of convolutions, the use of factorization was first proposed by Rigamonti et al. (2013). This decomposition can either be implicit Chollet (2017), using separable convolutions for instance (Jaderberg et al., 2014), or explicit, e.g using CP (Astrid and Lee, 2017; Lebedev et al., 2015) or Tucker (Kim et al., 2016) decompositions. These methods all fit within a more general framework of decomposition of the kernels of the decomposition, where the full kernel is expressed in factorized form, and the full, dense convolutional operation is replaced with a sequence of smaller convolutions with the factors of the decomposition (Kossaifi et al., 2020). Here, in contrast, we propose to factorize the **domain**, not the kernel, which allows us to perform **parallel** global convolution while remaining computationally tractable. The advantages include parallelism and better numerical stability, since we do not chain many operations. The factorization of the domain can lead to efficient computation, but the challenge is to find an explicit representation of the domain (Sec. 3.2).

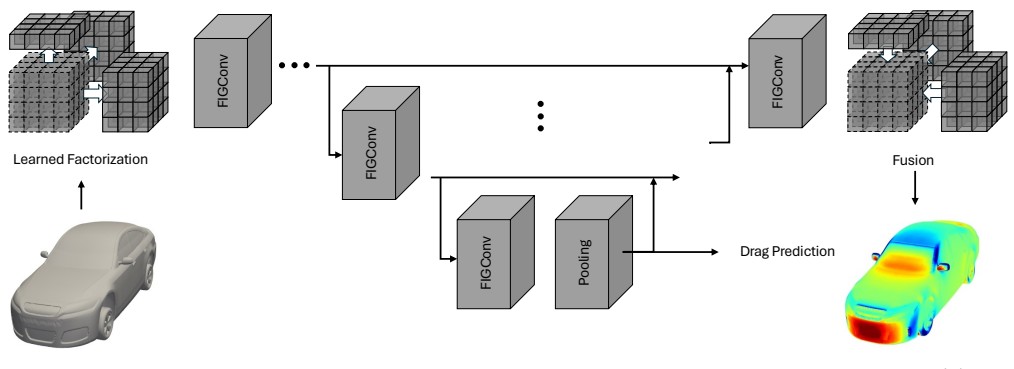

Figure 1: **FIGConvNet: ConvNet for drag prediction using FIG convolution blocks**. The encoder and decoder consist of a set of FIG convolution blocks and we connect the encoder and decoder with skip connections. The output of the encoder is used for drag prediction and the output of the decoder is used for pressure prediction.

## 3 FACTORIZED IMPLICIT GLOBAL CONVNET

In this section, we introduce our factorized implicit global convolution and discuss how we create implicit factorized representations, reparametrize the convolution, implement global convolution, and fuse the implicit grids. We then present a convolution block using factorized implicit grids and build a U-shaped network architecture for pressure and drag coefficient prediction. An overview diagram is provided in Fig. 1.

## 3.1 FACTORIZED IMPLICIT GRIDS

Our problem domain resides in 3D space with an additional channel dimension, represented mathematically as $\mathcal{X} = \mathbb{R}^{H_{\max} \times W_{\max} \times D_{\max} \times C}$ with high spatial resolution. Explicitly representing an instance of the domain $X \in \mathcal{X}$ is extremely costly in terms of memory and computation due to its

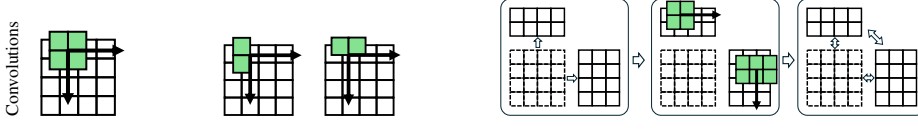

(a) Convolution     (b) Separable Convolution (Chollet, 2017)     (c) Factorized Implicit Global Convolution

Figure 2: From left to right, we have a regular convolution, a separable convolution, and our proposed factorized implicit global (FIG) convolution. Regular Convolution: Requires $O(N^2k^2)$ computation and the convolution kernel is not global. Separable Convolution: Involves a **sequence** of $O(N^2k)$ convolutions, but the convolution kernel is still not global. FIG Convolution: Requires $O(Nk)$ computation in **parallel**, with convolution kernels that are global in one axis in the respective factorized domain.

large size. Instead, we propose using a set of factorized representations $\{F_m\}_{m=1}^M$, where $M$ is the number of factorized representations. Each $F_m \in \mathbb{R}^{H_m \times W_m \times D_m \times C}$ has different dimensions, collectively approximating $X(\cdot) \approx \hat{X}(\cdot; \{F_m\}_{m=1}^M)$. These $\{F_m\}_{m=1}^M$ serve as implicit representations of the explicit representation $X$, and we refer to each $F_m$ as a factorized implicit grid throughout this paper.

Mathematically, we use MLPs to project features from the factorized implicit grids $\{F_m\}_m$ to the explicit grid $X$:

$$X(v) \approx \hat{X}(v; \{F_m\}_m, \theta) = \sum_m^M f(v, F_m; \theta_m) \tag{1}$$

$$f(v, F_m; \theta_m) = \sum_{i=i_v}^{i_v+1} \sum_{j=j_v}^{j_v+1} \sum_{k=k_v}^{k_v+1} \text{MLP}(F_m[i, j, k], v; \theta_m), \tag{2}$$

where $(i_v, j_v, k_v)$ is the smallest integer grid coordinate closest to the query coordinate $v \in \mathbb{R}^3$ and $\theta_m$ is the parameters of the MLP, which takes the concatenated features from the implicit grid $F_m$ and position encoded $v$ as an input.

To efficiently capture the high-resolution nature of the explicit grid $X$, we propose making one axis of $F_m \in \mathbb{R}^{H_m \times W_m \times D_m}$ low resolution. For example, $F_1 \in \mathbb{R}^{4 \times W_{\max} \times D_{\max}}$ where $H_{\max} \gg 4$ and $F_2, F_3$ to have low resolution $W$ and $D$ respectively. Thus, the cardinality of $X$, $|X|$ is much greater than that of the factorized grids, $|X| \gg \sum_m |F_m|$. Formally, this low-resolution size is the *rank r* of our factorized grid. For example, the $F_x \in \mathbb{R}^{r_x \times W_{\max} \times D_{\max}}$, $F_y \in \mathbb{R}^{H_{\max} \times r_y \times D_{\max}}$, and $F_z \in \mathbb{R}^{H_{\max} \times W_{\max} \times r_z}$. In experiments, since we use 3D grids, the rank is a tuple of 3 values, that we will denote as $(r_x, r_y, r_z)$, to represent the low resolution components of $(F_x, F_y, F_z)$. In practice, we will use $r_i < 10$ in place of $H_{\max}, W_{\max}, D_{\max} > 100$ thus making the cardinality of factorized grids $|F_m|$ orders of magnitude smaller than that of $|X|$.

Note that when we use a rank of 1, i.e. $(r_x, r_y, r_z) = (1, 1, 1)$, we have an implicit representation that resembles the triplane representation proposed in Chan et al. (2022) and Chen et al. (2022). This is a special case of factorized implicit grids that are used for reconstruction without convolutions on the implicit grids, fusion (Sec. 3.4), or U-shape architecture (Sec. 3.6).

## 3.2 FACTORIZED IMPLICIT CONVOLUTION

In this section, we propose a convolution operation on the factorized implicit grids. Specifically, we use a set of 3D convolutions on the factorized grids in parallel to approximate the 3D convolution on the explicit grid. Let $N$ be the dimension of the high-resolution axis and $K$ be the kernel size $N \gg K$. Then, the computational complexity of the original 3D convolution is $O(N^3K^3)$ and the computational complexity of the 3D convolution on the factorized grids is $O(MN^2K^2r)$, where $r$ is the dimension of the low-resolution axis, $M$ is the number of factorized grids. Mathematically, we have:

$$Y = \text{Conv3D}(X; W) \approx \sum_m Y_m = \sum_m^M f(\text{Conv3D}(X_m; W_m); \theta_\text{m}) \tag{3}$$

Figure 3: **Factorized Implicit Global Convolution 3D**: The FIG convolution first creates a set of voxel grids that factorizes the domain. This allows representing a high resolution voxel grid domain implicitly that can be computationally prohibitive to save explicitly. Then, a set of global convolution operations are applied in parallel to these voxel grids to capture the global context. Finally, the voxel grids are aggregated to predict the output.

where $Y$ and $\hat{Y}$ are the output feature maps of the original and approximation, and $W$ and $W_m$ are the weights of the original and factorized implicit convolutions.

### 3.3 EFFICIENT GLOBAL CONVOLUTION THROUGH 2D REPARAMETERIZATION

Large convolution kernels allow output features to incorporate broader context, leading to more accurate predictions (Peng et al., 2017; Huang et al., 2023). Experimentally, we find larger kernel sizes yield higher accuracy on the test set (Tab. 2). However, large kernel sizes can be impractical due to their computational complexity, which increases cubically with respect to the kernel size. To enable a larger receptive field without making computation intractable, we propose a 2D reparameterization of 3D convolution that allows us to apply large convolution kernels while maintaining low computational complexity. Specifically, we can reparameterize the 3D convolution to 2D convolution by flattening the an axis with channel. Mathematically, the 3D convolution on the flattened feature map is equivalent to 2D convolution with shifted kernel weights:

$$Y_m(i,j,k,c_o) = \sum_{c_{in}}^{C} \sum_{i',j',k'}^{K} X_m(i+i',j+j',k+k',c_{in})W(i',j',k',c_{in},c_o) \tag{4}$$

$$= \sum_{s=1}^{CK} \sum_{i',j'}^{K} X(i+i',j+j',k+\left\lfloor \frac{s}{C} \right\rfloor, s \mod C)W_m(i',j',\left\lfloor \frac{s}{C} \right\rfloor,c_o) \tag{5}$$

However, as we increase the kernel size $K \geq 2r-1$ where $r$ is the chosen rank, controlling the dimension of the low-resolution axis, we can reparametrize the convolution kernel into a matrix and replace the convolution with a matrix multiplication with the flattened input. For example, we can define a 1D convolution with kernel size $K=3$ and the axis of size 2 $(x_0, x_1)$ as:

$$\begin{bmatrix} y_0 \\ y_1 \end{bmatrix} = \begin{bmatrix} x_0 & x_1 & 0 \\ 0 & x_0 & x_1 \end{bmatrix} \begin{bmatrix} w_0 \\ w_1 \\ w_2 \end{bmatrix} \qquad \text{1-D spatial convolution with 1 channel} \tag{6}$$

$$= \begin{bmatrix} w_0 & w_1 \\ w_1 & w_2 \end{bmatrix} \begin{bmatrix} x_0 \\ x_1 \end{bmatrix} \qquad \text{reparametrization to 0-D space 2-vector matmul} \tag{7}$$

Using this reparametrization, we can convert a $D$ dimensional convolution with large kernels to $D-1$-dimensional convolution with $C \times N_D$ channels where $C$ is the original channel size and $N_D$ being the cardimality of the flattened dimension. This reparametrization does not change the underlying operation but reduces the practical computational complexity by removing redundant operations such as padding, truncation, and permutation involved in the 3D convolution. In addition, the kernel that is being flatten is global along the low-dimension axis as $K \geq 2r-1$. Experimentally, we find that the larger convolution kernels outperform smaller convolution kernels. However, if we do not use the reparametrization technique, the computation burden of the extra operations can outweight the added benefit (Tab. 2). Lastly, we name the final reparametrized convolution on the factorized implicit grids, factorized implicit global convolution (FIG convolution) as we apply global convolution on the factorized grids.

### 3.4 FUSION OF FACTORIZED IMPLICIT GRIDS

The convolution operation on the factorized implicit grids produces a set of feature maps $\{Y_m\}_m$ that in combination can represent the final feature map $\hat{Y}$ of one 3D convolution that approximates $Y$, which we do not explicitly represent. Thus, if we apply the factorized implicit global convolution multiple times on the same factorized implicit grids, there would be no information exchange between the factorized representations. To enable information exchange between the factorized representations, we fuse the factorized representations after each convolution by aggregating features from the other factorized grids. Mathematically, we use trilinear interpolation to sample features from $M - 1$ factorized grids $\{F_{m'}\}_{m' \neq m}$ and add the sampled features to the target grid $F_m$ by sampling from the all voxel locations $v_{ijk}$ of $F_m$. We visualize the final 3D convolution operation in Fig. 3.

### 3.5 LEARNING IMPLICIT FACTORIZATION

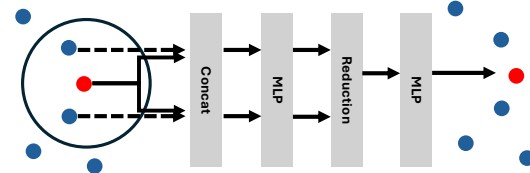

Figure 4: **Point Convolution**: The features from source and target nodes as well as offset are fed into an MLP to lift the features, which are then aggregated and projected back to the original feature space using an MLP.

We discussed how we perform global convolution on the factorized implicit grids. In this section, we discuss how we initialize the factorized implicit grids from an input 3D point clouds or a mesh. The traditional factorization of a large matrix of size $N$ requires $O(N^3)$ computational complexity, where $A \approx \hat{A} = P^T Q$. However, this decomposition is not ideal for our case where the resolution of the domain is extremely high. Instead, we propose to learn the factorized implicit grids from the input point clouds or meshes rather than first converting to the explicit grid $X \in \mathbb{R}^{H_{\max} \times W_{\max} \times D_{\max} \times C}$ – where $H_{\max}, W_{\max}, D_{\max}$ are the maximum resolutions of the domain and $C$ is the number of channels – and then factorize. We define a hyper paramter the number of factorized implicit grids $M$, as well as the size of the low-resolution axis $r$ and create $M$ factorized grids with different resolutions $\{F_m\}_m^M$, each with a different resolution $F_m \in \mathbb{R}^{H_m \times W_m \times D_m \times C}$. Then, we use a continuous convolution on each voxel center $v_{m,ijk}$ of $F_m$ to update the feature of the voxel $f_{m,ijk}$ from as set of features $f_n$ on point $v_n$ of the point cloud. Note that the input mesh is converted to a point cloud where each point represent a face of a mesh. We use $(i, j, k)$ to represent voxels and $n$ to indicate points:

$$f_{m,ijk} = \text{MLP}\left(\sum_{n \in \mathcal{N}(v_{ijk})} \text{MLP}(f_n, v_n, v_{ijk})\right), \quad \mathcal{N}(v, \Sigma) = \{i \|\Sigma^{-1/2}(v_i - v)\| < 1\} \quad (8)$$

where $\mathcal{N}(v, \Sigma)$ is the set of points around $v$ within an ellipsoid $(v_i - v)^T \Sigma^{-1}(v_i - v) < 1$ with covariance matrix $\Sigma \in \mathbb{R}^{3 \times 3}$ that defines the ellipsoid of neighborhood in physical domain. We use an ellipsoid rather than a sphere since the factorized grids have rectangular shape due to one low resolution axis. Each mlp before and after the summation use different parameters. To ensure the efficiency of the ellipsoid radius search, we leverage a hash grid provided by the GPU-acceleration library Warp (Macklin, 2022) and the pseudo-code is available in the Appendix.

### 3.6 UNET FOR PRESSURE AND DRAG PREDICTION

We combine factorized implicit global convolution with 2D reparameterization, fusion, and learned factorization to create a U-shaped ConvNet for drag prediction. While drag can be directly regressed using a simple encoder architecture, the number of supervision points is extremely small compared to the number of parameters and dataset size. Therefore, we add per-face pressure prediction as additional supervision, which is part of the ground truth since CFD simulation requires per-face pressure for drag simulation. We use the encoder output for drag prediction and the decoder output for pressure prediction. The architecture is visualized in Fig. 1.

## 4 IMPLEMENTATION DETAILS AND TRAINING

We implement all baseline networks and FIG convnet using pytorch. In this section, we describe the implementation details of the FIG convnet and the training procedure.

### 4.1 EFFICIENT RADIUS SEARCH AND POINT CONVOLUTION

One of the most computationally intensive operation in our network is the radius search in Eq. 8 for which we leverage a hash grid to accelerate the search. We first create a hash grid using Warp (Macklin, 2022) with the voxel size as the radius. Then, we query all 27 neighboring voxels for each point in the point cloud and check if the point is within a unit sphere. For non spherical neighborhoods, we scale the point cloud by the inverse of the covariance matrix $\Sigma$ and check if the point is within the unit cube.

We save the neighborhood indices, and number of neighbors per point in a compressed sparse row matrix format (CSR) and use batched sparse matrix multiplication to perform the convolution in Eq. 8. We provide a simple example of the radius search in the supplementary material.

### 4.2 FACTORIZED IMPLICIT GLOBAL CONVOLUTION

To implement 3D global convolution using factorized representations, we use minimum three factorized grids with one low resolution axis. We first define the maximum resolution of the voxel grid that can represent the space explicitly e.g. $512 \times 512 \times 512$. Then, we define the low resolution axis as $r_i$ for each factorized grid. Note that $r_i$ can be different for each factorized grid. For example, $512 \times 512 \times 2$, $512 \times 3 \times 512$, and $4 \times 512 \times 512$.

### 4.3 TRAINING PROCEDURE AND BASELINE IMPLEMENTATION

We train all networks using Adam optimizer with a learning rate of $10^{-3}$, step a learning rate scheduler with $\gamma = 0.1$ and step size of 25 epochs, and batch size of 16 for 100 epochs on NVIDIA A100 80G GPUs. We use a single A100 if batch size of 16 fits inside the memory and use 2 GPUs with batch size 8 each if not to make sure all experiments follow the same training configuration. The total training takes approximately 16 hours with two GPUs. For pressure prediction, we first normalize the pressure as all units are in the metric system and range widely. We denote $\bar{P}$ as the normalized pressure where it has 0 mean and 1 standard deviation. For both pressure prediction and drag prediction, we use the same mean squared error as the loss function. Training loss is simply the sum of both: $(\hat{c}_d - c_d)^2 + \frac{1}{N}\sum_i(\hat{\bar{P}}_i - \bar{P}_i)^2$ where $\hat{\cdot}$ denotes the prediction of $\cdot$ and $\bar{P}_i$ indicates the normalized pressure on the $i$-th face and $N$ the number of faces. We use the same training procedure, loss with the same batch size, learning rate, and training epochs for all baseline networks to ensure fair comparison. There are many representative baselines, so we chose an open-source framework that supports a wide range of network architectures and is easy to implement new networks. Specifically, we use the OpenPoint library (Qian et al., 2022) to implement PointNet segmentation variants, DGCNN, and transformer networks. We provide the network configuration yaml files in the supplementary material.

### 4.4 CODE RELEASE

We have publicly released all implementations of our FIG convolution, FIGConv U-Net architecture, and experiment configurations as part of the industry standard [HIDDEN FOR DOUBLE BLIND REVIEW] package, where users can download and preprocess public computational fluid dynamics datasets as well as train various neural network models.

## 5 EXPERIMENTS

We evaluate our approach using two automotive computational fluid dynamics datasets, comparing it with strong baselines and state-of-the-art methods: **DrivAerNet** (Elrefaie et al., 2024): Contains $4k$ meshes with CFD simulation results, including drag coefficients and mesh surface pressures. We adhere to the official evaluation metrics and data split. **Ahmed body**: Comprises surface meshes with approximately $100k$ vertices, parameterized by height, width, length, ground clearance, slant angle, and fillet radius. Following (Li et al., 2023), we use about $10\%$ of data points for testing. The inlet velocity ranges from $10m/s$ to $70m/s$, which we include as an additional input to the network.

Table 1: **Performance on on DrivAerNet**: we evaluate drag coefficient $c_d$ Mean Squared Error (MSE), Mean Absolute Error (MAE), Max Absolute Error (Max AE), coefficient of determination ($R^2$) of drag coefficient ($c_d$) prediction and inference time on the official test set. We evaluated the inference time on A100 single GPU. [†] numbers from the authors.

| Model | $c_d$ **Mean SE** ($\downarrow$) | $c_d$ **Mean AE** ($\downarrow$) | $c_d$ **Max AE** ($\downarrow$) | $c_d$ $R^2$ ($\uparrow$) | **Time (sec)** ($\downarrow$) |
|---|---|---|---|---|---|
| PointNet++ (Qi et al., 2017b) | 7.813E-5 | 6.755E-3 | 3.463E-2 | 0.896 | 0.200 |
| DeepGCN (Li et al., 2019) | 6.297E-5 | 6.091E-3 | 3.070E-2 | 0.916 | 0.151 |
| MeshGraphNet (Pfaff et al., 2020b) | 6.0E-5 | 6.08E-3 | 2.965E-2 | 0.917 | 0.25 |
| AssaNet (Qian et al., 2021) | 5.433E-5 | 5.81E-3 | 2.39E-2 | 0.927 | 0.11 |
| PointNeXt (Qian et al., 2022) | 4.577E-5 | 5.2E-3 | 2.41E-2 | 0.939 | 0.239 |
| PointBERT (Yu et al., 2022) | 6.334E-5 | 6.204E-3 | 2.767E-2 | 0.915 | 0.163 |
| DrivAerNet DGCNN (Elrefaie et al., 2024) [†] | 8.0E-5 | 6.91E-3 | **8.80E-3** | 0.901 | 0.52 |
| FIGConvNet (Ours) | **3.225E-5** | **4.423E-3** | 2.134E-2 | **0.957** | **0.051** |

Table 2: **Comparing Convolution Kernel Size (local and global) on DrivAerNet Normalized Pressure ($\bar{P}$) Prediction**: we evaluate Mean Squared Error (MSE), Mean Absolute Error (MAE), Max Absolute Error (Max AE), of normalized pressure and the coefficient of determination ($R^2$) of drag coefficient and inference time on the official test set. The local convolution suffers from long inference time. $(r_x, r_y, r_z) = (4, 4, 4)$ and kernel size $K \geq 2r-$ is global. (Sec. 3.3)

| Kernel size | $\bar{P}$ **Mean SE** ($\downarrow$) | $\bar{P}$ **Mean AE** ($\downarrow$) | $\bar{P}$ **Max AE** ($\downarrow$) | $c_d$ $R^2$ ($\uparrow$) | **Time (sec)** ($\downarrow$) |
|---|---|---|---|---|---|
| $3 \times 3 \times 3$ | 0.046845 | 0.11895 | 5.95431 | 0.93 | 0.054 |
| $5 \times 5 \times 5$ | 0.046364 | 0.11489 | 5.7173 | 0.943 | 0.061 |
| $7 \times 7 \times 7$ (Global) | 0.044959 | **0.1124** | **5.6795** | 0.955 | 0.079 |
| $7 \times 7 \times 7$ (2D Reparametrization) | **0.043818** | 0.11285 | 5.73351 | **0.957** | **0.051** |

## 5.1 EXPERIMENT SETTING

The car models in both datasets consist of triangular or quadrilateral meshes with faces and pressure values defined on vertices for the DrivAerNet and faces on the Ahmed body dataset. As the network cannot directly process a triangular or quadrilateral face, we convert a face to a centroid point and predict pressure on these centroid vertices for the Ahmed body dataset.

To gauge the performance of our proposed network, we considered a large number of state-of-the-art dense prediction network architectures (e.g., semantic segmentation) for comparison including Dynamic Graph CNN (DGCNN) (Wang et al., 2019), PointTransformers (Zhao et al., 2021), PointCNN (Qi et al., 2017a; Li et al., 2018), and geometry-informed neural operator(GINO)(Li et al., 2023). For the DrivAerNet dataset, we follow the DrivAerNet (Elrefaie et al., 2024) and sample $N$ number of points from the point cloud and evaluate the MSE, MAE, Max Error, and the coefficient of determination $R^2$ of drag prediction. For the Ahmed body dataset, we follow the same setting as (Li et al., 2023) and evaluate the pressure prediction.

## 5.2 RESULTS ON DRIVAERNET

Table 1 presents the performance comparison of various methods on the DrivAerNet dataset. Our FIGConvNet outperforms all state-of-the-art methods in drag coefficient prediction while maintaining fast inference times. PointNet variants (e.g., PointNet++, PointNeXt) perform well compared to transformer-based networks like PointBERT, likely due to the dataset's small size. For all baselines except DrivAerNet DGCNN, we incorporate both pressure prediction and drag coefficient prediction losses.

We analyzed the impact of convolution kernel size on pressure prediction (Table 2). Larger kernels approach global convolution but lead to performance saturation and slower inference. Our reparameterized 3D convolution achieves comparable performance with improved speed.

Figure 5b visualizes ground truth vs. predicted drag coefficients, demonstrating the network's ability to capture the distribution accurately. Figure 5a shows the effect of sample point count on prediction accuracy, revealing robustness across a wide range but potential overfitting with very high point counts. Qualitative pressure predictions are shown in Figure 6.

To assess the impact of factorized grid dimensions, we varied grid sizes (Table 3). Larger grids improved pressure prediction accuracy but degraded drag coefficient determination ($c_d \mathbf{R}^2$) and increased inference time.

Lastly, we removed the feature fusion between factorized grids proposed in Sec. 3.4. We observe that having no fusion in FIG convolution degrades performance but the gap is smaller when the grid $(r_x, r_y, r_z)$ are bigger. This suggests that while fusion remains important, its significance decreases with increasing grid size.

Table 3: **Choosing the rank: impact of the choice of on DrivAerNet on performance**: we evaluate normalized pressure $\bar{P}$ Mean Squared Error (MSE), Mean Absolute Error (MAE), Max Absolute Error (Max AE), coefficient of determination ($R^2$) of drag coefficient $c_d$ and inference time on the official test set. We trained for only 50 epochs for this experiment. Note that the car is facing +x axis and is the longest while -z is the gravity axis and is the shortest. See Sec. 3.1 for $(r_x, r_y, r_z)$ definition.

| $(r_x, r_y, r_z)$ | $\bar{P}$ **Mean SE** ($\downarrow$) | $\bar{P}$ **Mean AE** ($\downarrow$) | $\bar{P}$ **Max AE** ($\downarrow$) | $c_d$ $R^2$ ($\uparrow$) | **Time (sec)** ($\downarrow$) |
|---|---|---|---|---|---|
| (1, 1, 1) | 0.05278 | 0.1275 | 5.8266 | **0.9328** | **0.0305** |
| (3, 2, 2) | 0.05199 | 0.1250 | 5.8284 | 0.9249 | 0.0396 |
| (5, 3, 2) | 0.05131 | 0.1223 | 6.2350 | 0.8735 | 0.0399 |
| (10, 6, 4) | 0.05079 | **0.1221** | 5.6506 | 0.9243 | 0.0493 |
| (10, 10, 10) | **0.04999** | 0.1254 | **5.5343** | 0.8926 | 0.0610 |

Table 4: Impact of the factorized grid fusion (Sec. 3.4) on DrivAerNet: we evaluate normalized pressure $\bar{P}$ Mean Squared Error (MSE), Mean Absolute Error (MAE), Max Absolute Error (Max AE), coefficient of determination ($R^2$) of drag coefficient $c_d$, and inference time on the official test set. We trained for 50 epochs for this experiment. For no communication rows, we set the fusion layer in Sec. 3.4 to be identity and kept all the rest of the network the same.

| $(r_x, r_y, r_z)$ | $\bar{P}$ **Mean SE** ($\downarrow$) | $\bar{P}$ **Mean AE** ($\downarrow$) | $\bar{P}$ **Max AE** ($\downarrow$) | $c_d$ $R^2$ ($\uparrow$) | **Time (sec)** ($\downarrow$) |
|---|---|---|---|---|---|
| (3, 2, 2) | **0.05199** | **0.1250** | **5.8284** | **0.9249** | 0.0396 |
| (3, 2, 2) No Fusion | 0.053455 | 0.12683 | 6.28512 | 0.90413 | **0.0361** |
| (5, 3, 2) | **0.05131** | **0.1223** | 6.2350 | 0.8735 | **0.0399** |
| (5, 3, 2) No Fusion | 0.052921 | 0.12287 | **6.02101** | **0.88638** | 0.0451 |

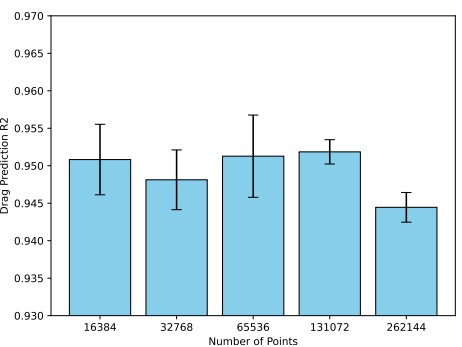
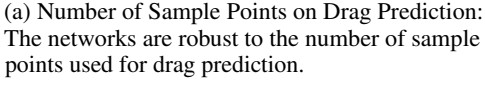
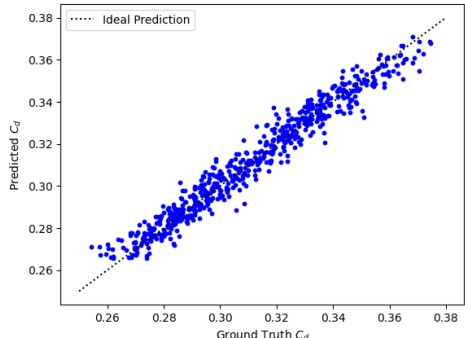

(a) Number of Sample Points on Drag Prediction: The networks are robust to the number of sample points used for drag prediction.

(b) Drag prediction vs. Ground truth drag on DrivAerNet. The drag prediction closely matches the drag ground truth with $R^2$ of 0.95.

## 5.3 Results on Ahmed body

Table 5 compares our method's performance on the Ahmed body dataset with state-of-the-art approaches Li et al. (2023), reporting normalized pressure MSE and model size. While GINO outperforms UNet and FNO, it achieves only $9\%$ pressure error. In contrast, our method attains a significantly lower normalized pressure error of $0.89\%$ with a smaller model footprint.

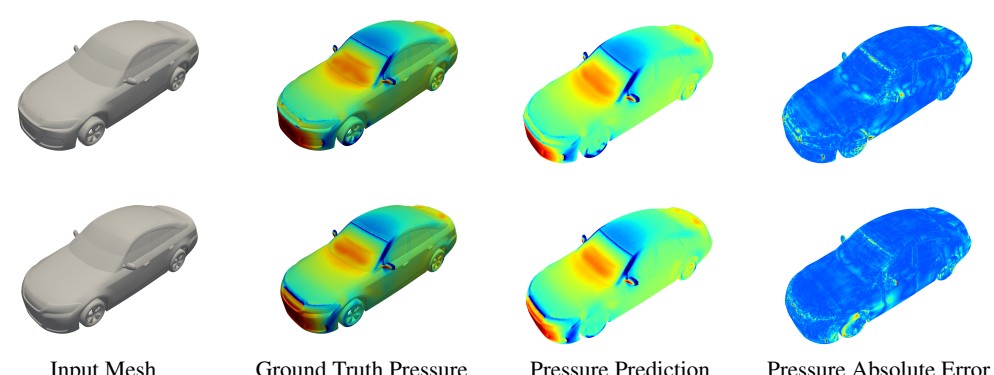

| Input Mesh | Ground Truth Pressure | Pressure Prediction | Pressure Absolute Error |

Figure 6: **Normalized Pressure Prediction and Error Visualization on DrivAerNet.** Our network predicts both drag coefficients and per vertex pressure. We visualize the ground truth pressure and prediction along with the absolute error of the pressure. Note that the pressures are normalized to highlight the errors clearly.

Table 5: **Ahmed Body Per Vertex Pressure Prediction Error** measured the normalized L2 pressure error per vertex on the test set. The top three rows are from Li et al. (2023).

| Model | Pressure Error | Model Size (MB) |
|---|---|---|
| UNet (interp) | 11.16% | 0.13 |
| FNO (interp) Li et al. (2020a) | 12.59% | 924.34 |
| GINO Li et al. (2023) | 9.01% | 923.63 |
| FIGConvNet (Ours) | 0.89% | 68.29 |

We further analyze the impact of grid resolution on network performance (Table 6). Our approach demonstrates robust pressure prediction across a wide range of grid resolutions, even with small grids. However, we observe that very high grid resolutions lead to overfitting on training data, resulting in decreased test performance.

## 6 CONCLUSION AND LIMITATIONS

In this work, we proposed a deep learning method for automotive drag coefficient prediction using a network with factorized implicit global convolutions. This approach efficiently captures the global context of the geometry, outperforming state-of-the-art methods on two automotive CFD datasets. On the DrivAerNet dataset, our method achieved an $R^2$ value of 0.95 for drag coefficient prediction, while on the Ahmed body dataset, it attained a normalized pressure error of 0.89%.

However, our approach has some limitations. The FIG ConvNet directly regresses the drag coefficient without incorporating physics-based constraints, which could lead to overfitting and poor generalization to unseen data. Additionally, our method is currently limited to the automotive domain with a restricted model design, potentially limiting its applicability to other fields. Looking ahead, we plan to address these limitations and further improve our model. Future work will focus on incorporating physics-based constraints such as Reynolds number and wall shear stress to enhance generalization.

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

## A  APPENDIX

## B  DATASETS

The foundation of CFD in the automotive industry provides insight into design and engineering. Prior comprehensive texts provide a solid overview of computational methods in fluid dynamics and dedicate a comprehensive overview of traditional CFD techniques (Ferziger et al., 2019) along with specification in automotive aerodynamics, also instrumental in understanding the principles (Katz, 2016). Solvers, such as OpenFOAM, a GPU-accelerated open-source solver, along with commercialized licensed solvers are widely used for solving CFD equations in automotive simulations (Jasak et al., 2007).

Such simulations consist of two main components, i) the car designs, complex geometry often developed special software, and ii) running large scale computation to solving multivariate coupled equations. Significant advancements have been enabled by the Ahmed body shape (Ahmed et al., 1984), a generic car model simple enough to enable high-fidelity industry-standard simulations while retaining the main features characterizing the flow of modern cars. Since, attempts have been made to improve the realism of the shapes. Shape-net (Chang et al., 2015) in particular has provided a valuable resource for simple car CFD simulations (Umetani and Bickel, 2018). Extending on Ahmed's body setting, the DrivAer data set introduces more complex and realistic car geometries (Heft et al., 2012), with subsequent efforts, producing large-scale aerodynamics simulation on such geometries(Varney et al., 2020). On such dataset, prior work attempts to predict car surface drag coefficients directly by bypassing the surface pressure prediction, pioneered by Jacob et al. (2021). However, this approach deploys an architecture applied to 3D voxel grids, forcing the method to scale only to low-resolution 3D grids version of the data. The lack of resolution obfuscates the fine details of geometry, making the network predict the same results for cars with different information. This is in contrast to our work that predicts pressure fields on large scales and detailed meshes.

**Ahmed body** consists of generic automotive geometries (Ahmed et al., 1984), simple enough to enable high-fidelity industry-standard simulations but retaining the main features characterizing the flow of modern cars. It was generated and used in the prior studies (Li et al., 2023) and contains simulations with various inlet velocities.

Ahmed body dataset, generated using vehicle aerodynamics simulation on the Ahmed body shapes (Ahmed et al., 1984), consists of steady state simulation of OpenFOAM solver on 3D meshes each with $10M$ vertices parameterized by height, width, length, ground clearance, slant angle and fillet radius. The dataset is generated and used in the prior studies (Li et al., 2023) and contains GPU accelerated simulations with surface mesh sizes of $100k$ on more than 500 car geometries, each taking 7-19 hours. We follow the same setting of this study using $10\%$ of shapes testing. The dataset is proprietary from NVIDIA Corp. Following this work, both of the deployed datasets are in the process of being made publicly available for further research.

Table 6: **Ahmed body Controlled Experiment** We vary the grid resolution and kernel size for analysis. $(r_x, r_y, r_z)$ is $(6, 2, 2)$ (Sec. 3.1) e.g. The three grid resolutions we used for the first three rows are 6×280×180, 560×2×180, 560×208×2.

| Max Resolution | Kernel Size | Pressure Error | Model Size (MB) |
|---|---|---|---|
| 560×208×180 | 3 | 3.40% | 105.0 |
|  | 7 | 3.31% | 417.1 |
|  | 11 | 2.56% | 979.7 |
| 280×104×90 | 3 | 2.89% | 105.0 |
|  | 7 | 3.05% | 417.1 |
|  | 11 | 2.93% | 979.7 |
| 140×42×45 | 9 | 1.65% | 667.29 |
|  | 11 | 2.59% | 979.7 |

**DrivAerNet** datasets is the parametric extension of DrivAer datasets. DriveAer cars geometries are more complex, real-world automotive designs used by the automotive industry and solver development (Heft et al., 2012), Fig **??**. Solving the aerodynamics equation for such geometries is a challenging task, and GPU-accelerated solvers are used to provide fast and accurate solvers, generating training data for deep learning purposes (Varney et al., 2020; Jacob et al., 2021). To train

our model on the DrivAer dataset, and to demonstrate the applicability of our approach to real-world applications, we use industry simulations from Jacob et al. (2021). DrivAerNet with 50 parameters in the design space. The dataset constitutes of 4000 data points generated using Reynolds-Averaged Navier-Stokes (RANS) formulation on OpenFoam solver on 0.5M mesh faces.

## B.1 BASELINE NETWORK CONFIGURATIONS

We list the network configurations used in the experiment in the appendix. We use OpenPoint Qian et al. (2022) for the baseline implementation and, with the configuration, you can specify the network architecture.

## B.2 BASELINE IMPLEMENTATIONS

We use an open source 3D point cloud library OpenPoint (Qian et al., 2022) to implement Point-Net++ (Qi et al., 2017b), DeepGCN (Li et al., 2019), AssaNet (Qian et al., 2021), PointNeXt (Qian et al., 2022), and PointBERT (Yu et al., 2022). In this section, we share the network architecture configuration used in the experiment.

```
NAME: BaseSeg
encoder_args:
  NAME: PointNet2Encoder
  in_channels: 3
  width: null
  strides: [2, 4, 1]
  mlps: [[[64, 64, 128]],
         [[128, 128, 256]],
         [[256, 512, 512]]]
  layers: 3
  use_res: False
  radius: 0.05
  num_samples: 32
  sampler: fps
  aggr_args:
    NAME: 'convpool'
    feature_type: 'dp_fj'
    anisotropic: False
    reduction: 'max'
  group_args:
    NAME: 'ballquery'
  conv_args:
    order: conv-norm-act
  act_args:
    act: 'relu'
  norm_args:
    norm: 'bn'
decoder_args:
  NAME: PointNet2Decoder
  fp_mlps: [[128, 128], [256, 128], [512, 128]]
```

Listing 1: **PointNet++ Configuration**

```
NAME: BaseSeg
encoder_args:
  NAME: DeepGCN
  in_channels: 3
  channels: 64
  n_classes: 256
  emb_dims: 256
  n_blocks: 14
  conv: 'edge'
  block: 'res'
  k: 9
  epsilon: 0.0
  use_stochastic: False
  use_dilation: True
  dropout: 0
  norm_args: {'norm': 'in'}
  act_args: {'act': 'relu'}
```

Listing 2: **DeepGCN Configuration**

```
NAME: BaseSeg
encoder_args:
  NAME: PointNet2Encoder
  in_channels: 3
  strides: [4, 4, 4, 4]
  blocks: [3, 3, 3, 3]
  width: 128
  width_scaling: 3
  double_last_channel: False
  layers: 3
  use_res: True
  query_as_support: True
  mlps: null
  stem_conv: True
  stem_aggr: True
  radius: [[0.1, 0.2], [0.2, 0.4], [0.4, 0.8], [0.8, 1.6]]
  num_samples: [[16, 32], [16, 32], [16, 32], [16, 32]]
  sampler: fps
  aggr_args:
    NAME: 'ASSA'
    feature_type: 'assa'
    anisotropic: True
    reduction: 'mean'
  group_args:
    NAME: 'ballquery'
    use_xyz: True
    normalize_dp: True
  conv_args:
    order: conv-norm-act
  act_args:
    act: 'relu'
  norm_args:
    norm: 'bn'
decoder_args:
  NAME: PointNet2Decoder
  fp_mlps: [[64, 64], [128, 128], [256, 256], [512, 512]]
```

Listing 3: **AssaNet Configuration**

```
NAME: BaseSeg
encoder_args:
  NAME: PointNextEncoder
  blocks: [1, 2, 3, 2, 2]
  strides: [1, 4, 4, 4, 4]
  width: 64
  in_channels: 3
  sa_layers: 1
  sa_use_res: True
  radius: 0.1
  radius_scaling: 2.5
  nsample: 32
  expansion: 4
  aggr_args:
    feature_type: 'dp_fj'
  reduction: 'max'
  group_args:
    NAME: 'ballquery'
    normalize_dp: True
  conv_args:
    order: conv-norm-act
  act_args:
    act: 'relu' # leakrelu makes training unstable.
  norm_args:
    norm: 'bn'  # ln makes training unstable
decoder_args:
  NAME: PointNextDecoder
```

Listing 4: **PointNeXt Configuration**

```
NAME: BaseSeg
encoder_args:
  NAME: PointViT
  in_channels: 3
  embed_dim: 512
  depth: 8
  num_heads: 8
  mlp_ratio: 4.
  drop_rate: 0.
  attn_drop_rate: 0.0
  drop_path_rate: 0.1
  add_pos_each_block: True
  qkv_bias: True
  act_args:
    act: 'gelu'
  norm_args:
    norm: 'ln'
    eps: 1.0e-6
  embed_args:
    NAME: P3Embed
    feature_type: 'dp_df'
    reduction: 'max'
    sample_ratio: 0.0625
    normalize_dp: False
    group_size: 32
    subsample: 'fps' # random, FPS
    group: 'knn'
    conv_args:
      order: conv-norm-act
    layers: 4
    norm_args:
      norm: 'ln2d'
decoder_args:
  NAME: PointViTDecoder
  channel_scaling: 1
  global_feat: cls,max
  progressive_input: True
```

Listing 5: **PointBERT Configuration**

## B.3 FIGCONVNET CONFIGURATION

We share the network configuration used in FIGConvNet experiments in the Appendix. The code will be released upon acceptance, and the network configuration below uniquely defines the architecture.

## B.4 FIG CONVNET ARCHITECTURE DETAILS

In this section, we provide architecture details used in our network using the configuration files used in our experiments.

```
num_levels: 2
kernel_size: 5
hidden_channels:
  - 16
  - 32
  - 48
num_down_blocks: [1, 1]  # defines the number of FIGConv blocks
    per hierarchy in encoder
num_up_blocks: [1, 1]  # defines the number of FIGConv blocks per
    hierarchy in decoder
resolution_memory_format_pairs:  # defines the grid resolutions
  - [  5, 150, 100]
  - [250,   3, 100]
  - [250, 150,   2]
```

Listing 6: **FIGConvNet Configuration**

## B.5   WARP-BASED RADIUS SEARCH

Algorithm 1 describes how we efficiently find the input points within the radius of a query point in parallel. It follows the common three step computational pattern in GPU computing when encountering dynamic number of results: Count, Exclusive Sum and Allocate, and Fill. We achieve excellent performance by leveraging NVIDIA's Warp Python framework, which compiles to native CUDA and provides spatially efficient point queries with its hash grid primitive.

---

**Procedure 1** GPU-accelerated points in a radius search

---

**Input:** input points $p$, query points $q$, radius $r$
**Output:** Results Array, Result Offset
  **procedure** COUNTRADIUSRESULTS(query points, input points, radius $r$)
      Step 1: Count number of results
      **for all** query points $q$ **do**
         **while** candidate $p \leftarrow$ hash-grid query($q$,$r$) **do**
            **if** $\|q - p\| <$ radius **then** count[q]++
            **end if**
         **end while**
      **end for**
  **end procedure**
  **procedure** COMPUTEOFFSET(count)
      offset $\leftarrow$ exclusive-sum(count)
      total number results $\leftarrow$ offset[last]
      results-array $\leftarrow$ alloc(total number results)
  **end procedure**
  **procedure** FILLRADIUSRESULTS(query points, input points, radius $r$, offset)
      **for all** query points $q$ **do**
         q-count $\leftarrow 0$
         **while** candidate $p \leftarrow$ hash-grid query($q$,$r$) **do**
            **if** $\|q - p\| <$ radius **then**
               results-array[offset[q-count]] $\leftarrow p$
               q-count++
            **end if**
         **end while**
      **end for**
  **end procedure**
  **procedure** POINTSINRADIUS(input points, query points, radius)
      count $\leftarrow$ COUNTRADIUSRESULTS(query points, input points, radius)
      offset, allocated results array $\leftarrow$ COMPUTEOFFSET(count)
      results array $\leftarrow$ FILLRADIUSRESULTS(query points, input points, radius, offset, results array)
  **end procedure**

---

