# OpenReview forum: "Factorized Implicit Global Convolution for Automotive Computational Fluid Dynamics Prediction"
_ICLR.cc/2025/Conference — Submitted to ICLR 2025_

### Official Review · Reviewer_aWtP · 2024-10-16

**Soundness:** 2
**Presentation:** 2
**Contribution:** 2
**Rating:** 5
**Confidence:** 4

**Summary:**

The paper introduces FIGConv, a U-shaped architecture to solve large-scale 3D CFD problems for the automotive industry, specifically to perform regression of the drag coefficient and the pressure field of a CFD simulation. FIGConv main contribution is the reduced computational complexity, of O(N^2), as opposed to most other pipelines with complexity O(N^3 logN^3). The authors correctly stress the main limitation of deep learning models employed in CFD, i.e. their scalability when processing large meshes.

 The reduced computational complexity of FIGConv is achieved through an approximation of high-resolution meshes via Factorised Implicit Grids, which significantly reduces the number of input elements by a factor of 10. This is done by creating M implicit grids via an MLP (learnt representation), each implicit grid having one low resolution axis. Such grids are then processed by Factorised Implicit Convolutions, which simplify 3D convolutions. FIC works by projecting 3D convolutions onto 2D ones by flattening the low resolution dimension of the FIGs.

Such choice of representation is based on recent advancements that use multiple orthogonal 2D planes to represent 3D quantities and on the technique of weights factorisation, which is however applied at domain level rather than at kernel level.

FIGConv is validated on two datasets, DrivAerNet and Ahmed body dataset and authors present results on estimated pressure field and drag coefficients. These two quantities are estimated at decoder and encoder stage, respectively, of the proposed architecture.

**Strengths:**

The paper introduces a representation for large 3D meshes and an approximation of 3D convolutions for the estimation of physical quantities in CFD simulations.

It tackles an important issue common to all learning-based PDE solvers, i.e. the handling of large meshes.

Good contextualisation with respect to previous work.

Results are promising and a comparison with multiple models is offered, especially in terms of speed.

The paper is fairly clear and easy to read.

**Weaknesses:**

Major points:

My main issue in the paper is Table 5, which shows some shortcomings in the overall approach:

1)  Reporting normalised error at test stage is formally incorrect, since at inference time we wish to have an estimate of actual pressure values, not normalised ones.

2) An error of 0.89% seems optimistic, especially considering that FIGConv is a convolutional neural network, while Neural Operators have been already demonstrated to be superior to most architectures when it comes to PDE modelling. The lack of figures and discussion on this result is also an important limitation. Li et al. use 0 mean, unit standard deviation normalisation, which yields a relative L2 norm error a bit higher compared to the denormalised version (you can do the math yourself and compare the normalised and denormalised relative L2 norm), which would mean a denormalised error similar or even lower compared to 0.89%. In the appendix you mention the same type of normalisation procedure, but, without further elements to assess the results and based on personal experience, an error of 0.89% is more likely to be deriving from a min max normalisation instead, since it squeezes the range of pressure values and significantly underestimates the error.

3) Intuition on why such good results are obtained on the Ahmed Body dataset but not on the DrivAerNet dataset is also missing.

The paper often cites GINO, by Li et al. The computational complexity cited by Li and by the authors of this paper, however, is not consistent, and I don’t see a complexity of O(N^3) reported by Li et al. I invite the author to be clearer on the meaning of N in their work.

The paper does not discuss the complexity added by the chosen representation: the MLP, the sampling and the radius search all play a role in the FIGConv pipeline and they have an impact on the computational complexity. Such impact, even if minor, is not mentioned.

There is no intuition behind why the proposed approach would work better - FIGConv is used to justify the reduced computational complexity (obtained by moving from a 3D to an almost-2D approach), the faster processing and the smaller model size, which I understand. The authors, however, do not provide a reason why it would actually perform better compared to neural operators, for example.

Results are listed but they are not thoroughly discussed.

Minor points:

Notation is not clearly introduced (e.g. meaning of N, M).

 Fig. 6 has no color bars, making it hard to interpret.

Some minor typos spotted: u-shaped vs U-shaped (lines 86, 109), Cardimality (line 262), ConvNet vs convnet (line 327).

General comments:

While I find the idea of representation fairly original, since it is an important question common to most research in large-scale, 3D PDE solver, I would have liked to see authors make a better argument on why FIGCov performs so well on the regression of the proposed physical quantities. I find the explanation of FIGCov working principle and benefits in terms of computational complexity clear and satisfactory, but those don't necessarily justify the enhanced accuracy. Do I think that FIGs are an original idea and worth presenting? Yes. Is it a game-changing approach in the field or would I use it in my own works? Probably not.
While the introduction of the work seems promising, the second half of the paper feels rushed and not that well executed, especially in the way in which results are presented, discussed and justified.

**Questions:**

Line 56: What does N represent in your paper? It is mentioned that GINO by Li et al. (NeurIPS 2023) has complexity O(N^3 logN^3), but from their paper, the reported complexity is O(N log N + N degree), with N, quoting, being “the number of mesh points” and “degree” stemming from the graph representation.

Line 160: What does the number of channels $C$ represent?

Line 194: How have $r_x, r_y, r_z$ been picked?

The reported computational complexity of O(N^2 logN^2) derives from the approximated 3D convolutions and the FIG representation. However, the complexity of the MLP which learns the FIGs, of the radius search of Eq. 8 and of the fusion of the feature maps is not discussed. How do those play a role, if any?

Fig. 6: What is the range of the of the pressure field? Is It normalised or denormalised? It is hard to tell how predictions compare with ground truth, also because the colors maps are not exactly consistent.

Table 5: The error metrics for Unit, FNO and GINO is relative L2 norm, but in the caption you mention L2 pressure error. Which one are you using, relative or absolute? I assume you meant relative?

Line 485: You report a normalized pressure error of 0.89%. Is this correct? A decrease of about 8% in error is huge, especially on a hard dataset like Ahmed body, and such improvement should be justified, which you don’t do in the text. Also, there is no figure on how the estimated pressure fields for the Ahmed body dataset look like to back up the claim. A discussion on this is especially needed since FIGConv performs only marginally better compared to other architectures on DrivAerNet, but significantly better on Ahmed body: what's the intuition behind such disparity between the two datasets?

 Moreover, a few points to be made:
 1) errors at test stage have to be reported on denormalised data, since what we care about in the end is the value of pressure in its original unit of measurement.    2) How are you normalising the data?

3) The fact that FIGConv is purely convolutional requires an explanation: why does GINO, which is a GNO + FNO architecture, perform poorly compared to FIGCov?

line 526: physics-based constraints can be hard to include, for example PINN are a whole different class of models. There are methods that provide an easy inductive bias, for example by embedding physical quantities within Clifford Algebra (cfr. Brandstetter et al., 2022, Ruhe et al. 2023), which could be an easier next step to boost the performance of FIGConv, especially since you are estimating two physical quantities at once. Have you considered that?

**Details Of Ethics Concerns:**

No ethics concerns.

---

> ### Author Response · Authors · 2024-11-23
>
> We sincerely thank the reviewer for their time and effort in actually reading this paper thoroughly, appreciating our original idea, and providing constructive comments. We will address the concerns in the following points.
>
> **W1 (Table 5), Q5 (Fig. 6), Q6 (Table 5)**: We used the same evaluation code as Li et al. The GINO authors can confirm this. We followed the same 0 mean Gaussian normalization, and the error of 9.01% from Li et al. and our 0.89% error used the same standardized dataset, loader, and evaluation code. We will explicitly state this in the revision.
>
> **W2 (GINO complexity)**: It uses 3D Fourier transformation (FFT), which is known to have $O(N^3 \log^3 N)$ complexity (https://en.wikipedia.org/wiki/Fast_Fourier_transform). The grid size in both ours and theirs is $N$.
>
> **W3 (radius search), Q4**: The radius search is a fairly expensive operation, and you are right to point it out. Normally, it requires $O(\text{radius}^3)$ complexity, but we used hash grid (L314), which requires $O(\|F\|)$ for the hash grid initialization where $\|F\|$ is the cardinality of the factorized grid $F$ (L193-L200), and insertion and search would require $O(1)$ per operation. Since $O(\|F\|)$ is $O(N^2)$ complexity, we can say that our radius search is $O(N^2)$ complexity. However, note that this radius search is done only on (references) voxels to points (neighbours) since we need to project features from points to grids.
>
> **W4 (intuition)**: FIGConv works better because of the scalability of the factorized representation. Scaling a neural network to support larger, higher resolution, deeper networks has been the holy grail in the field of computer vision and natural language processing, and we highlight that this aspect can be incorporated into the field of computational fluid dynamics.
>
> **W4 (neural operators)**: Neural operators have identitcal network architectures (convolution, normalization, activations, etc). What makes the neural operators operators is how the metric spaces are preserved so that the learned operations are functional, which have been standard in 3D perception tasks.
> We did not mention anything about the operators as we can explain the same network without involving the additional complexity of explaining the neural operators (Occam's razor).
> In short, the underlying networks used in neural operator are the same convnets used in other 3D perception tasks and CFD is just another dense prediction task.
>
>
> **Q1 (L56)**: [GINO](https://github.com/neuraloperator/neuraloperator/blob/202426c28fc33dc7ac61511eb9cc9cba6a8e3cbd/neuralop/models/gino.py#L273) consists of 1. input point convolution from point to 3D grid, 2. Fourier Neural Operator, 3. and then another point convolution from 3D grid to point.
>
> The FNOBlock calls [SpectralConv](https://github.com/neuraloperator/neuraloperator/blob/202426c28fc33dc7ac61511eb9cc9cba6a8e3cbd/neuralop/layers/spectral_convolution.py#L420) which 1. FFT on the data to transform data to frequency domain, 2. Elementwise multiply transformed data with weights. 3. IFFT to convert it back to the original domain.
> Among all these operations, the most expensive ones are FFT and IFFT which has the complexity of $O(N^3 \log N^3)$ where $N$ is the grid resolution.
>
> We have confirmed this with the GINO authors.
>
> **Q2 (L160)**: This is the transformed feature size like channel size of a layer in a convnet. We have input point cloud and features on points. We transform the point cloud to a grid using Eq.8. This results in a 3D voxel grid where each voxel has a $C$-vector as a feature. However, $\mathcal{X}$ is an explicit feature grid which costs $O(N^3)$ to store. Instead, we use $\{\mathcal{X}_m\}$, the factorized representations.
>
>
> **Q3 (L194)**: We ran controlled experiments on Tab.3 to determine the size of $r$s.
>
> **Q4 (radius search)**: See W3.
>
> **Q5, Q6**: See W1.
>
> **Q7 (L485), Q9 (why better than GINO)**: The GINO FNO consists of three layers. 1. Point convolution from point to grid. 2. FNO (FFT + elementwise multiplication + IFFT) this is one convolution according to the [convolution theorem](https://en.wikipedia.org/wiki/Convolution_theorem). 3. Point convolution from grid to point. Three layer convnet is a very shallow network. However, we used a deep U-Net thanks  to the strong scalability of the factorized representation. However, the DGCNN used by DrivAerNet paper is a very deep network similar to ours. We will clarify this in the revision.
>
> **Q8 (normalization)**: See W1.
>
> **Q9**: see Q7
>
> **Q10 (L526, physics-based constraints)**: Yes, essentially, most physics-based constraints use additional loss terms to penalize the violation. Our preliminary experiments show that these additional loss terms did not help the performance of GINO. Would be happy to discuss more after the rebuttal.

---

> > ### Comment · Reviewer_aWtP · 2024-11-25
> > **Rebuttal**
> >
> > I would like to thank the authors for the time taken in addressing my concerns. In terms of complexity, the authors have done a good job in clarifying their contribution. it is clear to me now the meaning of $N$ and where the cube term appears with the employed notation. The computational advantage of FIGConv has been clearly discussed.
> >
> > I still feel, however, that the reduced computational complexity does not necessarily justify the results reported. I don't feel that the authors have addressed in a satisfactory manner my concerns on the results on the Ahmed body dataset, and the fact that they have used the same code of GINO et al. is not a sufficient contextualisation of the obtained results.
> >
> > Authors in GINO report **normalised error for training and denormalised error for test**. If authors of FIGConv used the same evaluation code, then they also report denormalised test error, which is in contrast with what they state in the caption of Table 5. If that was a mistake in the manuscript, then it should have been noticed when pointed out and addressed with the rebuttal. To me, this imprecision alone is critical.
> >
> > I have to reiterate that 1 order of magnitude decrease in the error metrics is also a substantial claim. If that is the case, and I am positive it is, the addition that the code employed is the same to that of GINO, as mentioned in the rebuttal, is not a sufficient nor convincing enough explanation for me.
> >
> > I will wait to read the revised manuscript.

---

### Official Review · Reviewer_nryP · 2024-10-23

**Soundness:** 2
**Presentation:** 3
**Contribution:** 2
**Rating:** 5
**Confidence:** 4

**Summary:**

The author used factorized implicit global convolution as a supervised learning algorithm for CFD problems for the prediction of pressure quantities. They claimed that the complexity of their algorithm is O(N^2), compared to other methods, which have the same complexity of O(N^3). They tested their work over 3D benchmark problems. More specifically, they integrated factorized implicit global convolution with U-Nets. They used point convolution to convert grid vertices of unstructured meshes to the space of the factorized implicit global convolution networks. They also compared their model to others, such as PointNet++, PointNext, etc. Comparisons were made using different metrics such as the R2 score, root mean squared errors, and visual comparisons.

**Strengths:**

--> Well written, high-quality figures, 3D benchmark problems (rather than 2D), comprehensive error analysis,

**Weaknesses:**

-->The following statement from the manuscript is inaccurate and based on a false assumption. The literature was not thoroughly reviewed:

**Prior works in the deep learning literature, focusing on large-scale point clouds, ranges from the use of graph neural networks and point-nets to the u-shaped architectures along with advanced neighborhood search (Qi et al., 2017a; Hamilton et al., 2017; Wang et al., 2019; Choy et al., 2019; Shi et al., 2020). However, these methods make assumptions that may not be valid when applied to CFD problems.**

This is not true, PointNet and PointNet++ have been successfully applied to 2D and 3D problems in CFD.

--> The main issue of this work is grid projection, a challenge faced by all convolution-based neural networks. The authors use PointNetConv to map 3D finite volume vertices to a convolutional representation, but this mapping introduces errors when converting from unstructured grids to Cartesian grids. As a result, the advantages of PointNet and graph neural networks, which excel in handling unstructured data, have been overlooked, and not discussed by the authors.

--> Another wrong assumption by the authors. Please note that PointNet and PointNet++ have been used to predict the velocity and pressure of whole space in CFD problems not only the surface of objects. In this work, the authors only predict the pressure on the surface.

--> The literature review overlooks several researchers who have done significant work in applying PointNet and graph neural networks to CFD problems.

--> The paper seems a bit misleading, as the authors initially propose that their work addresses CFD problems, but they ultimately reduce the problem to predicting the pressure distribution on the surface. The introduction needs to be rewritten to reflect this more accurately.

--> In the end, I would say that the authors have done a lot of work; however, the proposed framework is very complicated and not scalable, and the results are not impressive.

**Questions:**

--> On page 15, line 808, the figure number is missed and we see ??. Please fix it.

--> I suggest writing the Navier-Stokes Equations, at least, in Appendix.

--> I suggest adding the visual results of the Ahmed Body benchmark in the Appendix.

--> The authors claim to have compared their model with others, such as PointNet++. In the appendix, they provided the code. On line 845, we see that the radius is set to 0.05. This is a very sensitive hyperparameter and plays a significant role in the performance of PointNet++. Did the authors fine-tune this parameter or not? When comparing our model to previous models, we should be fair and aim to achieve the best performance for all models involved.

--> I have also listed some issues in the Weaknesses section. Please address them.

---

> ### Author Response · Authors · 2024-11-23
>
> The majority of the weaknesses (4 out of 5) are due to 7-year-old architectures (PointNet and PointNet++) and why they should be performing better. We thoroughly compared more recent models such as DGCNN, GraphNet, PointBERT, and an architecture specifically designed for CFD (DrivaerNet DGCNN) along with PointNet++.
>
> We believe that the reviewer's criticism discards all recent developments in the field in favor of old architectures that have been proven to be less effective in 3D semantic segmentation, instance segmentation, etc., which are similar to dense prediction tasks like ours.
>
> Regarding the last weakness:
>
> > In the end, I would say that the authors have done a lot of work; however, the proposed framework is very complicated and not scalable, and the results are not impressive.
>
> We experimentally showed in Tables 1 and 5 that our framework is more scalable and achieved state-of-the-art performance while being 10x faster than the previous best. This seems like a very subjective rather than scientific criticism.
>
> For all minor questions, we will revise the paper.
>
> **Q4 (PointNet++ radius)**: The radius is indeed a sensitive hyperparameter, and we swept different values and chose 0.05 to be the largest radius that an A100 can support since radius search has high computational overhead and smaller radii result in lower performance. This also shows the worse scalability of PointNet++ and DGCNN.

---

> ### Comment · Reviewer_nryP · 2024-11-27
> **[Reply]**
>
> Thank you for addressing the questions and providing your responses. After reviewing the authors' answers, I have observed a lack of a thorough literature review in this work. The authors are working in the area of machine learning and CFD, where two distinct communities are actively contributing: one from computer science and the other from fluid dynamics.
>
> It is important to note that researchers in the fluid dynamics community typically publish their work in journals such as Journal of Computational Physics, Physics of Fluids, and Journal of Fluid Mechanics, rather than computer science conferences. As such, familiarity with both perspectives is crucial. While I agree with the authors that PointNet and PointNet++ may be considered outdated in computer graphics, their applications in CFD are still relatively novel, as evidenced by recent publications in the fluid dynamics community.
>
> Here are a few examples:
>
> https://papers.ssrn.com/sol3/papers.cfm?abstract_id=4944997 (**2024**)
>
> https://ieeexplore.ieee.org/document/10662040 (**2024**)
>
> https://www.sciencedirect.com/science/article/pii/S0045782524003864 (**2024**)
>
> https://www.sciencedirect.com/science/article/pii/S1000936122002795 (**2023**)
>
> https://www.sciencedirect.com/science/article/pii/S0021999122005721 (**2022**)
>
> https://doi.org/10.1063/5.0033376 (**2021**)
>
> ... and many others.
>
> After carefully reconsidering the manuscript and the authors' response, I have updated my score from 6 to 5.

---

### Official Review · Reviewer_9kRw · 2024-10-31

**Soundness:** 3
**Presentation:** 2
**Contribution:** 2
**Rating:** 3
**Confidence:** 4

**Summary:**

The authors introduce FIGConv, an efficient spatial representation learning method designed to address large-scale 3D simulation modeling challenges. Their experimental results are remarkable, showing significant improvements across two large datasets. Notably, on the Ahmed Body dataset, FIGConv achieves an order-of-magnitude improvement over GINO, a result that is indeed impressive. However, the manuscript does not provide sufficient insight into the specific aspects of the FIGConv architecture that contribute to such substantial performance gains compared to previous state-of-the-art models like GINO. Understanding what aspects of the proposed model structure enable these advancements would be beneficial.

**Strengths:**

1. The experimental results are impressive.
2. The authors introduce Factorized Implicit Grids and Factorized Implicit Convolution methods to the neural operator field as an efficient representation learning modeling method.

**Weaknesses:**

1. As far as I know, many attention-based methods deal with this problem, and representation learning for large-scale 3D data. Why did the author not discuss it in the related work section and conduct comparative experiments as the baseline in the methods section? Oformer, GNOT, Transolver.

2. The method part of the paper is poorly stated. First, what is the problem definition? What are the inputs and outputs of the model? Second, what are the core innovations of your model? How to apply the method of 3d decomposed representations in the cv domain to the operator domain? So what's the innovation of your model?

3. The experimental part, at present, seems to only have the main result part, which shows the effect of the proposed method on the two data sets. As I initially questioned, the model achieved impressive results, so why? Why such a big improvement? Which modules play an important role?

4. The authors did not open source or provide pseudocode for the algorithm, leaving us unclear on exactly how the method works.

**Questions:**

The experimental results of this paper are impressive, but the manuscript is unable to provide a strong explanation, and if the authors provide some convincing feedback, I will consider improving the score further.

1. In the related work section, can the author join the discussion of attention-based methods? Many attention-based methods show quasi-linear computational complexity, which is in sharp contrast to the quadratic complexity of the proposed method. Please refer to the author for further elaboration.

2. As suggested in the previous section, can the authors go into further detail in the methods section? What is the problem definition? What are the inputs and outputs of the model? After input mesh, how is learned factorization obtained, and in the final fusion, how is the corresponding pressure obtained? What is data streaming?

3. What are the core innovations of your model? How to apply the method of 3d decomposed representations in the cv domain to the operator domain? So what's the innovation of your model?

4. Can we add comparison experiments of attention-based methods in the experimental part? GNOT, Transolver, etc., have linear complexity.

5. Then 4, in the experimental section, can more ablation experiments be provided? As I initially questioned, the model achieved impressive results, so why? Why such a big improvement? Which modules play an important role?

6. The authors did not open source or pseudocode the algorithm, leaving us unclear on exactly how the method works.

---

> ### Author Response · Authors · 2024-11-23
>
> Thanks for the review. However, we have to mention that the rating is extremely unfair since the reasons for such low score is not citing your work that did not use the same datasets; claiming theres only main result when we provided 6 tables for ablation studies; and claiming we have not open sourced when we specifically mentioned that we already released the code.
>
> **W1 (transformer related works)**: The paper specifically tackles large scale 3D automotive data, as our title suggests. OFormer and GNOT deal with small scale
> problems not related to automotive. Transolver tackles this but was not peer-reviewed
> before submission. Also, it used 790 car shapes with 32k mesh points, which are
> small compared to the DrivAerNet dataset and Ahmed body dataset.
>
> The PointBERT is an attention-based model that has been used in 3D perception tasks and we have compared with our models in Table 1.
>
> However, we will add the papers you mentioned in the revision.
>
> **W2 (problem definition), Q3**: The problem is defined in L160. However, due to the
> datasets we used, we did not make a clear distinction on the outputs. The DrivAerNet
> dataset uses drag coefficient as the output as the main metric, whereas prior
> works used pressure prediction on the car mesh as the main metric. We will clarify
> this in the revision.
>
> **W3 (experimental results), Q6**: We added the main result in Table 1. As well as
> controlled experiments in Table 2, 3, 4, and 6. Please specify which ablation study you would like to see.
>
> **W4 (open source) and Q7**: We already released the code on GitHub. Also, In Sec. 4.4 Code Release, we said
>
> > We have publicly released all implementations ... as part of the industry standard [HIDDEN FOR DOUBLE BLIND REVIEW] package
>
> Due to the double blind, we will reveal the link to the repository only to the area chair in confidential comments.
>
>
> **Q1 (if the authors provide some convincing feedback, I will consider improving the score further.), Q4**: Scalability of FIGConv is the key innovation. Let me remind you what our paper says:
>
> Introduction: The key innovation is the factorized implicit convolution (L67-78).
>
> > With Factorized Implicit Grids, we approximate high-resolution domains using a set of implicit grids, each with one lower-resolution axis
>
> Related Work: Many previous works failed to use high resolution meshes, grids, due to memory limitations. (L98-100)
>
> >  However, many of these networks exhibit poor scalability due to the cubic complexity of memory and compute O(N3) or slow neighbor search.
>
> Decomposed representations in 3D reduce complexity (L105-106)
>
> > Such representation significantly reduces the memory complexity of implicit neural networks on 3D continuous planes
>
> Method: We use factorized grids to lower the computation and memory complexity (L160-178)
>
> > Explicitly representing an instance of the domain X ∈ X is extremely costly in terms of memory and computation ...
>
>
> **Q2 (attention-based methods)**: Attention-based methods you mentioned do not have quasi-linear complexity. There is $O(N^2)$ complexity for the slice representation (or a plane). Within each slice, attention with $M$ tokens requires $O(M^2)$ complexity. Therefore, the overall complexity is $O(N^2 M^2)$ for a single slice. However, since there are $N$ slices, the overall complexity is $O(N^3 M^2)$. We will clarify this in the revision.
>
>
> **Q5 (comparison with attention-based methods)**: None of these methods used the DrivAerNet dataset or the Ahmed body dataset, so we cannot compare directly, but we can try.

---

> > ### Comment · Reviewer_9kRw · 2024-11-26
> > **Rebuttal**
> >
> > Thanks to the authors for their response. The author's response did not address my concerns and I decided to leave the score unchanged.
> >
> > As mentioned before, my core concern is that the authors claim very good results (e.g., an order of magnitude improvement on the Ahmed-body data), but from the ablation experiments the authors have shown, we cannot see from which module such a striking improvement was achieved. Why? Is it just Factorized Implicit Grids? What is behind it?

---

### Official Review · Reviewer_n5EV · 2024-11-03

**Soundness:** 1
**Presentation:** 2
**Contribution:** 2
**Rating:** 3
**Confidence:** 4

**Summary:**

This manuscript describes a method for regressing drag coefficient and pressure field for automotive fluid dynamics simulation data. The proposed method achieves higher accuracy with lower computational complexity by representing the data by a set of different projections and fusing the corresponding set of predictions. The resulting models are evaluated on the DrivAerNet and Ahmed body Datasets.

**Strengths:**

- the algorithm shows good performance on the chosen application
- the manuscript makes an attempt to generalize prior work that used plane projections or separable convolutions
- in most parts, the manuscript is easy to read

**Weaknesses:**

- the mathematical description is in several cases unclear or wrong, see questions below
- the manuscript is partly rushed an contains errors, see detailed issues
- the proposed method is designed for a very specific application, which limits the contribution, see final question below.
- detailed issues
    - Pfaff et al. 2020 a and b is the same
	- L055 girding
	- L301-305 after reading this part three times, it still does not reveal its meaning, needs reformulation.
	- table 2, caption a symbol is missing after the kernel size definition
	- figure 5 lost its number, says just a and b
	- L514: table 6 is in the appendix, not the main paper

**Questions:**

- L043: when it says "most of these works", it implies at least one doesn't. So which approach is that and what is different with that approach compared to the others?
- 2.1/3.1 why is the approach called "factorization"? There is no product in (1)/(2), just a linear combination of non-linear projections. What is the relation to the mathematical concept of factorization?
- for one position v, the complexity is reduced, but doesn't the regression task for the pressure fields require prediction at the full grid resolution, i.e. $O(N^3)$? Please elaborate more on the computational complexity, also in the case of computing the pressure field prediction as done in the experiments.
- is the function f() in (3) the same as in (1) and if so, what argument is replaced with the 3d convolution? If it is a different function, how is it defined?
- what is the difference between global convolution as used in 3.3/figure 2c and a simple scalar product / contraction alternatively a point-wise operation along the chosen dimension? What are the advantages of the identified differences?
- looking at the identity (4)=(5), a contradiction is obtained for e.g. incrementing $s$ from 1 to 2 vs $c_{in}$ from 1 to 2 (if we assume C>2). The kernel $W$ might have different values in (4) but the same value is used in (5). Or is there anything wrong with this example? Please provide a proof if possible.
- L252: 3D convolution is a linear operator, why wouldn't it be possible to rewrite in matrix multiplication with the flattened vector in all cases? What are the differences between the standard matrix formulation of convolution and the proposed formulation?
- (6) vs (7): it is well known that convolution is commutative. Correlation requires additional reflection. To which extent differs (7) from correlating with the swapped (and reflected) arguments?
- L301 how is a continuous convolution realized in the voxel-based grid? Which signal model and approximation is used?
- (8) is a function over $m$, obviously missing on the RHS, but where? Please provide a corrected/complemented formulation.
- L423 what is meant by DFCNN not incorporating both? Both are occurring table 1? Please revise the formulation or explain.
- L655 this reference is quite important for the submitted manuscript. Has this reference been published or is it still just a preprint?
- to which other application domains can the proposed method be applied to? Please discuss potential applications of the proposed method beyond automotive CFD and highlight any modifications that might be necessary for other domains.

---

> ### Author Response · Authors · 2024-11-23
>
> Most of the weaknesses and questions from n5EV were about simple algebra confusion, which we visualized to help the reviewer understand the basic math.
>
> It is unfortunate that the reviewer only caught up in the algebra and fails to see the idea of the paper, which is factorizing the domain (not the kernel) for accelerated 3D convolutions for scalable learning of the CFD prediction networks.
>
> Also, reviewer failed to capture the importance of how an algorithm works is implemented on a computer. It is as important if not more important than an idea itself if it cannot be implemented in practice. Our approach for the 2D reparametrization is mathematically identical to 3D convolution (L263-268), but we showed that it is experimentally very effective (Tab. 2).
>
> **W1 (unclear or wrong) I2 (L301-305 ... three times) and Q9 (L301 how is a continuous convolution realized) Q10 ((8) is a function)**:
> The confusion seems to arise from the continuous convolution. The input is a
> point cloud (L294,298,303,311), and we need to project features from continuous
> points to voxel centers. This is done through continuous convolution [Monte
> Carlo Convolution for Learning on Non-Uniformly Sampled Point Clouds, 2018;
> Lagrangian Fluid Simulation with Continuous Convolutions, 2020].
>
> Input: feature $f_n$ which is located on a continuous point (L303)
> Output: feature $f_{ijk}$ on voxel grid (L304)
>
> Equation 8 is an aggregation of all neighbor features and another linear transformation.
>
> The m could be added on MLP${}_m$ to make the context clear.
>
> The graph convolution or continuous convolution is not the main focus of the manuscript so we refer the reviewer to the references for more details.
>
> **W2 (rushed)**: We thank the reviewer for a thorough review. These are minor
> errors and will be corrected.
>
> **W3 (specific application)**: We thank the reviewer for the suggestion. The
> application is indeed specific, but we respectfully disagree that this is a
> weakness of the manuscript. In fact, our specific focus makes our contribution
> readily applicable to industrial applications.
>
> **Q2 (factorization)**:  We tried to answer this question in section 3.1 and
> 3.2. The factorization of the grid, here, is implicit, and equation (2) shows
> how to project from that grid back to the explicit (full) grid. In turn,
> equation (3) shows how to perform convolutions directly in the implicit grid
>
> **Q3 (complexity)**: The target task is pressure prediction on the SURFACE of a
> mesh, not volume of the entire space. Thus, the complexity is $O(N^2)$.
>
> **Q4 (function f)**: These are the same. f is an MLP that takes parameter $\theta$. (L190)
>
> **Q5 (global convolution), Q7 (rewrite matrix multiplication)**: Yes, all convolutions can be written as flattened matrix multiplication, which is what Sec. 3.3 says. In L263,
>
> > This reparametrization does not change the underlying operation but reduces the ... complexity by removing redundant operations ...
>
> In detail, convolution in cuDNN transforms data using im2col operation, and it is
> ineffective when we have an extremely skinny or fat matrix. Table 2 shows 3D
> convolution is slower due to this overhead. In the majority of cases, input data
> is not extremely skinny or fat, and cuDNN is more efficient than the flattened
> matrix multiplication.
>
> **Q6 (incrementing s)**
> $s$ is the row-major order [raveled
> index](https://numpy.org/doc/stable/reference/generated/numpy.ravel_multi_index.html)
> of the z axis and the channel axis. Simply put, a matrix is internally stored as
> one contiguous array on memory and indexing is done through unravelling the
> matrix indices into scalars.
>
> Let’s say we have one index s, with matrix size C1xC2, we can convert between array and matrix back and forth using `i,j = s // C2, s % C2`.
>
> In equation 5, we incorrectly switched mod and floor, it should be $W_m(... s \mod C, c_o)$. Thank you for pointing out.
>
> **Q8 (commutative)**: This is a misunderstanding. This is not a correlation matrix; it is simple algebra, and we visualized it in [Google Slides](https://docs.google.com/presentation/d/1buel66IzWnBVO-wxVRpAsktP-9o_3NB1vGdIzWioDLQ/edit?usp=sharing).
>
> **Q12** (Has this reference been published): NeurIPS 2023

---

> > ### Comment · Reviewer_n5EV · 2024-11-25
> > **rebuttal**
> >
> > The reviewer respectfully advises the authors to carefully read the instructions for preparing the rebuttal.
> >
> > Beyond that, the authors have the responsibility to clearly and correctly describe their work in their manuscript to make it readable to the relevant part of the conference audience, including experts in the area such as the assigned reviewers. Even if the authors present work on reasonable ideas and achieving good results, this does not replace the requirements for the manuscript regarding correctness and clarity.
> >
> > The use of signal processing notation and the lack of accurate mathematical concepts suggests that the authors have a different background and the reviewer's comments are an opportunity to correct the manuscript. The low assessment is due to the amount of issues and necessary modifications for meeting the quality requirements of the conference.
> >
> > Said that, the reviewer highly appreciates the approach as such as well as relevant implementation questions. However, it is important not to deviate from good practice regarding terminology (see "factorization"), accurate definitions of functions (f()), accurate description of the transition between paradigms (continuous vs discrete convolution), and correctness (Q8 referred to the correlation between two functions, not the correlation matrix - convolution (commutative) reflects the kernel before computing the scalar products).
> >
> > As only few of the questions were answered in a correct way, the reviewer keeps the assessment.

---

> > > ### Author Response · Authors · 2024-11-26
> > >
> > > Regarding your comment on the readability, other reviewer including n5EV (the reviewer) mentioned the paper is easy to read. The only difficulty seemed to arise from the reviewer's exposure to the practical field.
> > >
> > > The kernels we used in convolution are identical to the convolution used in signal processing as $g \ast h = \sum_i g_i h_{i-k}$. However, none of the practical papers use such reflection as this is simply identical to $h'_k = h_{-k}$ as it has no bearing on the practical implementation. The insistance on the reflection of the kernel is unheard of in the field as most readers need not be reminded of the basic definition of convolution.

---

> > > > ### Comment · Reviewer_n5EV · 2024-11-28
> > > >
> > > > The original review says "in most parts, the manuscript is easy to read", in case the authors have missed this. The reviewer's concern is not about literal presentation but about technical accuracy, which the reviewer considers insufficient for this venue.
> > > >
> > > > Regarding the convolution, the reviewer is well aware of the missing reflection in the literature. That was not the point of the original comment in the review: "(6) vs (7): it is well known that convolution is commutative. Correlation requires additional reflection. To which extent differs (7) from correlating with the swapped (and reflected) arguments?"
> > > >
> > > > The authors never replied to the actual question.

---

### Author Response · Authors · 2024-11-23

First of all, we are deeply concerned about the reviewer's lack of understanding and effort in reviewing this paper.

9kRw's rationale for such a low score is: 1. not citing their own work that did not use standardized large-scale datasets, 2. claiming a lack of experiments without any specifics when we provided 6 tables for ablation studies, and 3. claiming we have not open-sourced when in the paper we dedicated one section to how we have already open-sourced (the link was shared with ACs to maintain anonymity). This is very unprofessional and shows a lack of effort in reviewing the paper.

nryP's rationale for such a low score is that a 7-year-old architecture should perform better, which is based more on personal bias than objective analysis, as the field of 3D perception has progressed significantly since 2017. We have already compared our model with more modern architectures that are proven to be better in 3D perception tasks, and our model outperforms all of them.

Finally, n5EV's rationale for such a low score is that the math is unclear or wrong, when in fact, these are nitpicking minor math confusions that we have addressed in detailed comments in our respective rebuttals.

aWtP provided constructive comments and appreciated our original idea as:

> representation is based on recent advancements that use multiple orthogonal 2D planes to represent 3D quantities and on the technique of weights factorisation, which is however applied at domain level rather than at kernel level.

> Results are promising and a comparison with multiple models is offered, especially in terms of speed.

> Do I think that FIGs are an original idea and worth presenting? Yes.

---

### Meta-Review · Area_Chair_JZor · 2024-12-19

**Metareview:**

This paper on computational fluid dynamics received 4 highly critical reviews. While reviewers appreciated the excellent performance of the proposed method, they raised multiple serious issues:

- technical soundness,
- positioning and description of the relevant literature,
- framing of the contributions, missing identification and justifications of the gains,
- weak presentation.

The communication between the authors and the reviewers was difficult, with the authors accusing all reviewers of misunderstanding and wrongly evaluating their contributions.

The AC sides with the reviewers and judges that the responses to the reviewers' questions were handwavy and focused on details, avoiding essential issues. In the scientific peer review process, it is the responsability of the authors of a paper to convince the reviewers of the its validity and soundness, in particular when there is a consensus of 4 independent and knowledgeable reviewers on specific issues. In this particular case, there is an essential requirement for a scientific paper to link gains in experimental evaluation to methodological contributions, which the paper has failed to provide. The AC judges that the paper is promising but not ready.

**Additional Comments On Reviewer Discussion:**

The reviewers and authors engaged in discussions

---

### Decision · Program_Chairs · 2025-01-22

Reject